# Confident Adaptive Language Modeling

**Tal Schuster**[1,*]   **Adam Fisch**[2,*]   **Jai Gupta**[1]

**Mostafa Dehghani**[1]   **Dara Bahri**[1]   **Vinh Q. Tran**[1]   **Yi Tay**[1]   **Donald Metzler**[1]

[1]Google Research        [2]CSAIL, MIT

## Abstract

Recent advances in Transformer-based large language models (LLMs) have led to significant performance improvements across many tasks. These gains come with a drastic increase in the models' size, potentially leading to slow and costly use at inference time. In practice, however, the series of generations made by LLMs is composed of varying levels of difficulty. While certain predictions truly benefit from the models' full capacity, other continuations are more trivial and can be solved with reduced compute. In this work, we introduce Confident Adaptive Language Modeling (CALM), a framework for dynamically allocating different amounts of compute per input and generation timestep. Early exit decoding involves several challenges that we address here, such as: (1) what confidence measure to use; (2) connecting sequence-level constraints to local per-token exit decisions; and (3) attending back to missing hidden representations due to early exits in previous tokens. Through theoretical analysis and empirical experiments on three diverse text generation tasks, we demonstrate the efficacy of our framework in reducing compute—speedup of up to $\times 3$—while provably maintaining high performance.[1]

## 1 Introduction

Recent advances in Large Language Models (LLMs) have led to breakthroughs in language understanding and language generation across almost every widely-used Natural Language Processing (NLP) task considered in the field today [5; 15; 17; 20; 51; 52; 53; 75; 89; 73]. Autoregressive language modeling provides a flexible framework for solving complex tasks with a unified natural language input and output format, while also relaxing the need for large-scale task-specific data collection and training [67; 15; 17; 58; 80]. The large size of LLMs, however, results in massive computational load that might be limiting for certain real-world applications (e.g., machine translation) [9; 30; 42; 49; 59; 63; 71]. This is especially pronounced in the autoregressive decoding process where the full stack of Transformer layers is repeatedly computed for each output token [37; 40; 86].

While large models do better in general, the same amount of computation may not be required for every input to achieve similar performance (e.g., depending on if the input is easy or hard) [66]. Early exiting is a promising approach to decreasing the computational cost of multilayered architectures such as those used in Transformer-based LLMs, where the number of layers used by the model is dynamically decided on an input-by-input basis [18; 23; 57; 60; 70]. In this setting, an LLM can choose to generate a new token based off the representation at an intermediate layer instead of using the full model, and save computation as a result. A natural question that arises, however, is when is it a good decision to exit early, as opposed to wait? Naively choosing when to exit can be suboptimal in terms of saving computation time, and also result in unpredictable degradations to model performance, especially when predictions depend on each other, as in autoregressive language generation.

---

[*]Project leads. Correspondence to: talschuster@google.com
[1]Code: https://github.com/google-research/t5x/tree/main/t5x/contrib/calm

36th Conference on Neural Information Processing Systems (NeurIPS 2022).

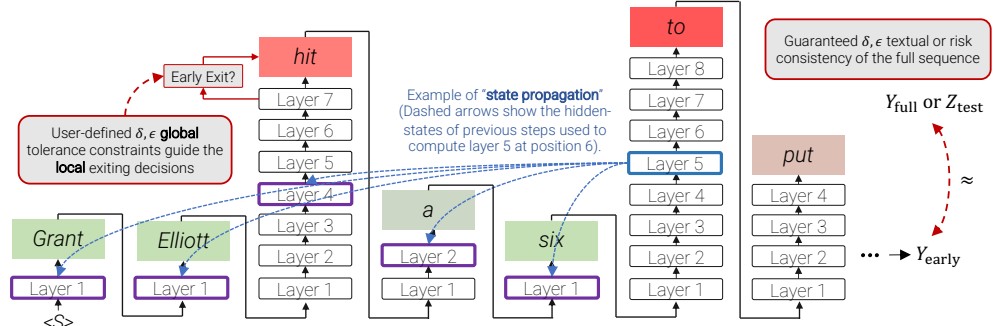

Figure 1: Illustration of CALM generation (see Figure 4 for the full example) with local per-token early exiting decisions that provably satisfy global user-defined constraints on the full sequence.

In this work, we analyze the early exiting paradigm for LLMs, and present a principled method for increasing model efficiency while remaining confident in the quality of the resulting predictions. Specifically, we develop a method for calibrating local, per-token, exit decisions such that global, sequence-level constraints—as determined by lexical or semantic sequence-level metrics like ROUGE or BLEURT score—are provably maintained with arbitrarily high probability (e.g., 95%). This process, which we call **C**onfident **A**daptive **L**anguage **M**odeling (CALM), is illustrated in Figure 1.

Our approach leverages recent techniques in distribution-free risk control in order to create confident generations with strong statistical guarantees [2; 3; 10]. Concretely, suppose we have been given a calibration set $\mathcal{S}_{\text{cal}} := \{P_i\}_{i=1}^n \in \mathcal{P}^n$ of independent and identically distributed (i.i.d.) prompts to our LLM (e.g., paragraphs to be summarized, sentences to be translated, or questions to be answered via language modeling). Let $P_{\text{test}}$ be a new i.i.d. test prompt to our LLM, where $Y_{\text{early}} := \text{LLM}_{\text{early}}(P_{\text{test}})$ and $Y_{\text{full}} := \text{LLM}_{\text{full}}(P_{\text{test}})$ are the adaptive and standard outputs of our LLM, respectively. In order to be satisfied with $Y_{\text{early}}$, we might require it to be *textually consistent* with $Y_{\text{full}}$. Given any bounded text dissimilarity function $\mathcal{D}$, we aim to calibrate the early-exiting LLM such that its predictions agree to a tolerance $\delta$ with the full model in expectation with high probability,

$$\mathbb{P}\Big(\mathbb{E}\big[\mathcal{D}(Y_{\text{early}}, Y_{\text{full}})\big] \leq \delta \,\big|\, \mathcal{S}_{\text{cal}}\Big) \geq 1 - \epsilon, \tag{1}$$

where the randomness is over draws of $\mathcal{S}_{\text{cal}}$, and $\epsilon \in (0, 1)$. Eq. (1) has the significant advantage of being achievable using only unlabeled calibration data $\mathcal{S}_{\text{cal}}$ (a quality that is critical for few-shot tasks, for example). Enforcing textual consistency with the original $Y_{\text{full}}$, however, may be unnecessarily strict for certain tasks, especially where multiple generations may be acceptable. As an alternative, given a calibration set of prompts paired with a set of (potentially multiple) target references, $\mathcal{S}_{\text{cal}} := \{(P_i, Z_i)\}_{i=1}^n \in (\mathcal{P} \times 2^{\mathcal{Y}})^n$, and any bounded risk function $\mathcal{R}$, we also consider an objective that enforces *risk consistency* by limiting the relative increase in risk of the predictions $Y_{\text{early}}$ compared to $Y_{\text{full}}$, with respect to the set of test-time references $Z_{\text{test}}$, i.e.,

$$\mathbb{P}\Big(\mathbb{E}\big[\mathcal{R}(Y_{\text{early}}, Z_{\text{test}}) - \mathcal{R}(Y_{\text{full}}, Z_{\text{test}})\big] \leq \delta \,\big|\, \mathcal{S}_{\text{cal}}\Big) \geq 1 - \epsilon. \tag{2}$$

Within the constraints of either Eq. (1) or Eq. (2), the goal of our work is to find the most computationally *efficient* $Y_{\text{early}}$, i.e., generations that exit as early as possible while still maintaining our desired performance guarantees. In order to achieve this, it is necessary to develop a reliable signal for how likely local, per-token early-exit decisions are to disrupt the global properties of the complete sequence. Here, we first analyze how errors are propagated in Transformer-based LLMs, and then present an effective and efficient scoring mechanism for assigning "consistent early-exit" confidence scores after each layer used during the generation of a new token. The decision to exit or not is based on these scores, and is carefully calibrated using $\mathcal{S}_{\text{cal}}$ such that our performance bounds are provably satisfied.

Finally, we empirically validate our method on multiple, diverse NLP generation tasks, including text summarization, machine translation, and question answering. Our experiments demonstrate the potential of CALM in reducing the average complexity of the model and accelerating inference by about $\times 3$ while reliably controlling for high performance.

**Contributions.** In summary, our main contributions are as follows:

- A framework (CALM) for reliably accelerating Transformer-based LLM generations.
- A systematic analysis of the token-wise early exit mechanism that motivates a simple-but-effective class of confidence measures and threshold functions that are used as part of the CALM framework.
- An empirical demonstration of CALM's efficiency gains on three diverse generation datasets.

## 2 Related Work

Improving inference-time efficiency of LLMs has been an ongoing effort of the research community over the past several years [49; 72; 85], leveraging techniques such as knowledge distillation [6; 32; 36; 69; 69; 78; 56], floating point quantization [71; 65], layer pruning [24], vector dropping [38], and others [41]. Another line of work involves conditional computation to train larger models that only use a sparser subset of the full network during inference, for example by routing over mixture-of-experts [9; 22; 39; 91], recurring modules [18; 29; 35], or accessing external memory [82]. These models, however, still use the same amount of compute for all input examples.

Here, we focus on *adaptive compute*, a specific kind of conditional compute that aims to dynamically allocate different computational power per example, with the goal of reducing the overall complexity while maintaining high performance. This approach, often referred to as *early-exiting* [16; 25; 47; 74; 79; 87], is complementary to many of the solutions above and can potentially be combined with them. Multiple early-exit techniques for encoder-only Transformers (e.g., BERT [20]) have been recently proposed [8; 34; 43; 44; 45; 60; 68; 83; 90; 92]. Most of these methods rely on intrinsic confidence measures (e.g., based on the softmax distribution), while others try to predict the routing in advance [46; 70], or train a small early-exit classifier [57; 84], as we also examine here. These measures can be calibrated to reliably guarantee consistency of the early prediction with the full model [57]. However, the techniques used for encoder-only classifiers are unsuitable for global consistency constraints with a sequence of dependent predictions, which are inherent in the decoding process of autoregressive language models, which we address here.

Our work is also motivated by recent findings on the existence of saturation events in LMs, where the top-ranked prediction is unchanged after some layer and is propagated upward. Geva et al. [28] examined interactions of the hidden-state with feed-forward layers to predict these events. However, they only consider local single predictions and do not address the challenges involved with sequence generation. Our early-exit LM architecture most closely relates to Elbayad et al. [23], who found a token-level early-exit classifier to provide the best efficiency-performance tradeoffs on machine translation. Here, we introduce a theoretically-grounded calibration method for provably controlling the quality of the full sequence. By doing so, we provide reliable efficiency gains—deriving local early exiting decisions from the global desirable constraints. Moreover, we introduce several model improvements and empirical analyses, including (1) analyzing the primary sources of performance degradation, leading us to propose a decaying threshold function for better tradeoff control without inflating the search space; (2) improving the early-exit classifier training; and (3) experimenting with two new tasks.

Our calibration procedure for connecting global constraints to local decisions, relates to recent research around distribution-free uncertainty quantification [1; 62; 77]. Several methods were developed in recent studies to expand and adjust the theoretical framework for obtaining practical efficiency gains on target applications [4; 7; 21; 26; 27; 48; 88]. Here, we frame our consistency requirements around the Learn then Test (LTT) framework [3], and leverage the approximately monotonic behavior of our confidence measures and the nested structure of our problem, that by definition guarantees consistency with large enough threshold, to form tight and effective bounds.

## 3 Early Exiting for Adaptive Language Modeling

In the following, we describe and analyze the early-exiting Transformer LM. We begin with a brief recap of the Transformer architecture (§3.1) and early exiting (§3.2) for convenience, following previous work [23; 70; 76]. We then investigate the effects of early exiting on model performance, and identify primary sources of performance degradation and how to alleviate them (§3.3)—which guide our architecture and training design (§3.4) and proposed per-token confidence measures (§3.5).

### 3.1 The Transformer architecture

We use the Transformer sequence-to-sequence model, based on the T5x implementation [55]. Here, we only review simplified details of the Transformer architecture relevant to early-exiting, and refer the reader to Vaswani et al. [76] for full details. At a high level, both encoder and decoder networks contain $L$ stacked layers, where each layer is composed of a multi-head self-attention sub-layer, followed by a feedforward sub-layer, each with residual connections and layer normalization. The decoder network has an additional multi-head attention sub-layer that attends to the encoder states.

Consider a prompt $x = (x_1, \ldots, x_p)$, processed by the encoder to yield encoder states $(e_1, \ldots, e_p)$, and the current, partially generated response $(y_1, \ldots, y_t)$. When generating the next token $y_{t+1}$, the decoder computes a decoder state $d_t^i$ for layer $i$ out of $L$ as:

$$h_t^i := \text{Attention}(d_t^{i-1}, d_{1:t-1}^{i-1}); \quad a_t^i := \text{Attention}(h_t^i, e_{1:p}); \quad d_t^i := \text{FeedForward}(a_t^i). \quad (3)$$

Internal to each of the attention mechanisms, written as $\text{Attention}(x, z_{1:m})$ for some input $x$ and sequence of $m$ states $z_{1:m}$, $x$ is first projected to a query vector $q := \mathbf{W}_Q x \in \mathbb{R}^{\dim_k}$, while $z$ is projected to a matrix of key-value vectors, $\mathbf{K} := \mathbf{W}_K z_{1:m} \in \mathbb{R}^{m \times \dim_k}$ and $\mathbf{V} := \mathbf{W}_V z_{1:m} \in \mathbb{R}^{m \times \dim_v}$. The output $o$ is then computed as $o := \text{softmax}\left(q\mathbf{K}^\top / \sqrt{\dim_k}\right) \mathbf{V}$.

Multi-head and normalization components are omitted for brevity. Each layer uses different projections $\mathbf{W}_Q^i$, $\mathbf{W}_K^i$, and $\mathbf{W}_V^i$ (which are also unique for computing $h_t^i$ versus $a_t^i$).

Finally, after layer $L$, a distribution over vocabulary tokens $y_{t+1} \in \mathcal{Y}$ is computed via a softmax-normalized linear classifier $\mathbf{W}_L$, where $p(y_{t+1} \mid d_t^L) = \text{softmax}(\mathbf{W}_L d_t^L)$.

## 3.2 Decoding with early exiting

Instead of always making a prediction based on the representation at the final layer, $d_t^L$, the key idea in early-exiting is to choose $y_{t+1}$ more quickly, if confident, by computing $p(y_{t+1} \mid d_t^i) = \text{softmax}(W_i d_t^i)$ for some intermediate layer $i < L$. Concretely, let $c_t^i \in [0, 1]$ denote some local confidence score for layer $i$ while processing token $t$, where higher values indicate a higher propensity to exit early (we will propose effective instantiations of $c_t^i$ in §3.5). Let $\lambda_t^i \in [0, 1]$ denote some local early-exiting threshold, where the model exits early if $c_t^i \geq \lambda_t^i$, or otherwise proceeds to compute the next representation, $d_t^{i+1}$. The (greedily chosen) prediction $y_{t+1}$ can then be written as:

$$y_{t+1} := \begin{cases} \arg\max p(y_{t+1} \mid d_t^1) & \text{if } c_t^1 \geq \lambda_t^1, \\ \arg\max p(y_{t+1} \mid d_t^2) & \text{if } c_t^2 \geq \lambda_t^2, \\ \quad \vdots & \\ \arg\max p(y_{t+1} \mid d_t^L) & \text{otherwise.} \end{cases} \quad (4)$$

Note that due to the self-attention mechanism of the Transformer, computing the input hidden state $h_t^i$ for layer $i$ depends on $d_{1:t-1}^{i-1}$, i.e., the output hidden states of the previous layer for *all* the tokens that have been generated so far.[2] Therefore, if the model has early exited at some layer $j < i - 1$ for a token $s < t$, then $d_s^{i-1}$ is not available. As an approximation, we set $d_s^k = d_s^j$ for all layers $k > j$ following Elbayad et al. [23], with the understanding that this will introduce some error. In the next section, in addition to other factors, we will analyze the impact of this copied state on performance.

## 3.3 The effects of early exiting on error propagation

We perform several controlled experiments to investigate the behavior and the potential of early-exiting during decoding. We use an 8-layer T5 encoder-decoder and the CNN/DM dataset for these experiments. See §5 for more details on this model and data.

### 3.3.1 State propagation

First, we control for the correctness of the predicted tokens to examine the effect of state copying (§3.2), and also measure an approximate upper bound for compute reduction. We use an *oracle* confidence measure that exits at the earliest layer that agrees with the top prediction (i.e., replacing the conditions in Eq. 4 with $\arg\max p(y_{t+1} \mid d_t^i) = \arg\max p(y_{t+1} \mid d_t^L)$). Hence, the only factor that can cause divergence in the generation is the state copying mechanism for skipped layers. The results of this experiment are highly encouraging. This oracle achieves an ROUGE-L score of 38.24, compared to 38.32 with the full model, while only using an average of 1.53 layers per token. We also try an oracle that always uses $d_{1:t-1}^1$ and it reaches 38.31 ROUGE-L. These results indicate that (1) the model is robust to state copying from lower layers, and (2) there is remarkable potential for saving compute—by up to ×5.2—while preserving performance, given a good confidence measure.

We also experiment with copying the projected states $\mathbf{K}^j, \mathbf{V}^j$ to skipped layers $k > j$. This version of the oracle results in a significant drop in performance to 23.02 ROUGE-L. Overall, we conjecture

---

[2] In autoregressive decoding, the $\mathbf{k}_s, \mathbf{v}_s$ vectors are cached to avoid repetitive compute for tokens $t > s$.

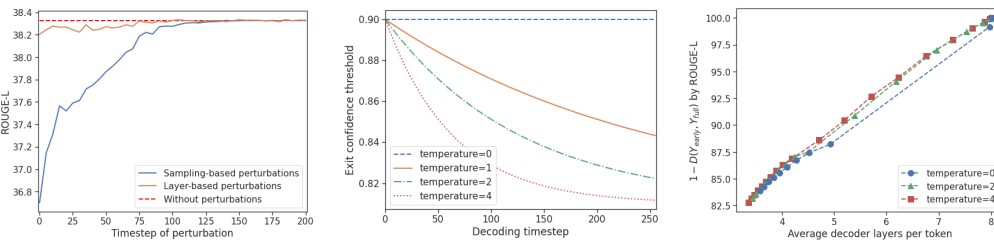

| (a) Sensitivity to local errors. | (b) Decaying threshold ($\lambda = 0.9$). | (c) Perf.-efficiency tradeoffs. |

Figure 2: Earlier noise in the decoding process has greater effect on the overall output (a), though in practice the affect of early exits is minor due to high performance of early layers. A decaying confidence threshold (b) allows finer control over the performance-efficiency tradeoff (c).

that the self-attention at layer $i$ for token $t$ can safely use hidden-states $d_s^j$ for $j < i - 1$ as key-values of tokens $s < t$, as long as the projections $\mathbf{W}_{K/V}^i$ of layer $i$ are used. Notably, this projection can now be computed concurrently for all skipped layers as they all use the same $d$ from the exited layer.

### 3.3.2 Sensitivity to local errors

Next, we examine the impact of local token modifications—which might occur due to early exits—on the whole generated sequence. We experiment with two kinds of perturbations: *sampling-based*, where we select the 10th-ranked token according to layer $L$; and *layer-based*, where we select the the first layer's prediction at timestep $t$. All other tokens are predicted greedily by layer $L$. As shown in Figure 2a, earlier perturbations result in lower sequence-level scores as there are more tokens that might suffer from the divergence. The degradation, though, is much smaller with layer- compared to sampling-based perturbations since, in practice, the early exit predictions are mostly accurate.

**Decaying threshold.** Following the above observation, we introduce a decaying early-exiting threshold that is more permissive towards exiting as the decoding process continues. Motivated by the logarithmic behavior in Figure 2a, we use an exponential function with a user-defined temperature $\tau$:

$$\lambda'(\lambda, t) := \text{clip}_{[0,1]}\left(\frac{9}{10}\lambda + \frac{1}{10}e^{-\tau \cdot t/N}\right), \tag{5}$$

where $N$ is the maximum output length. Figure 2b illustrates this function. Essentially, this function presents an effective compromise between simply using the same threshold for all tokens, and searching over a huge space of per-position different thresholds. Practically, it supports finer and better control over the performance-efficiency tradeoff compared to a single threshold. Figure 2c presents the outcomes of a search over $\lambda$ with steps of $0.01$ and softmax-based confidence (§3.5). With the single threshold variant ($\tau = 0$), attempting to improve the efficiency will lead to a drastic drop of more than 10 points in the textual similarity against the full model's prediction. In contrast, the decaying thresholds reveal several intermediate points with desirable tradeoffs to consider.

### 3.4 Training early exit classifiers for local consistency

While our goal is to preserve the quality of the complete output sequence, we note that this doesn't necessarily demand local token-level consistency. Consider the target sequence *"the concert was wonderful and long."* An output that switches the order of adjectives to *"the concert was long and wonderful"* would be called consistent by most semantic measures (and obtain 100 token-$F_1$ score). Yet, the sentences diverge at the first adjective *long* which is semantically different from *wonderful*.

Training for global consistency, however, could be challenging [81] as it depends on possibly noisy signals that might affect the learning, and also breaks the efficient teacher-forcing training strategy of LMs that relies on local-decisions. On the other hand, perfect local consistency implies global consistency. Therefore, we opt to train for local consistency, which requires minimal changes to the training procedure, and relax the local requirement to a global one during inference.

Specifically, similar to Elbayad et al. [23], we average losses for each layer to obtain the objective

$$\mathcal{L} = \sum_{i=1}^{L} \omega_i \mathcal{L}_i, \quad \text{where} \quad \sum_{i=1}^{L} \omega_i = 1. \tag{6}$$

$\mathcal{L}$ is the negative log-likelihood loss. We set $\omega_i = i/\sum_{j=1}^{L} j$ to favor higher layers, and find this objective to mostly preserve the full model's performance compared to regular training. We note that there is some misalignment between this training and inference behavior due to the hidden states of skipped layers. However, as discussed in §3.3.1, the performance is not affected if the hidden-state is copied.

### 3.5 Local confidence measures

We experiment with three confidence measures for Eq. (4) that differ in their parameter and compute operation efficiencies. Our experiments (§6) will also show that they differ in their predictive power.

**Softmax response.** We take the difference between the top two values of $\mathrm{Softmax}(\mathbf{W_i} d_t^i)$. With a large output vocabulary, this results in many floating point operations (FLOPs)—though, the next layer $i + 1$ can start its computation in parallel, avoiding additional runtime.

**Hidden-state saturation.** As a simple parameter-free and fast to compute alternative, we take the cosine similarity $\mathrm{sim}(d_t^i, d_t^{i-1})$ for $i > 1$. By definition, the first possible exit is at the second layer (unless $\lambda = 0$). This measure tries to identify early saturation events of the hidden-state [28].

**Early exit classifier.** We train a dedicated linear classifier $\mathcal{M}$ to predict the likelihood of exiting with local consistency given the current hidden-state: $c_t^i = \mathcal{M}(d_t^i)$. This measure is very fast to compute at inference, and adds only $|d| + 1$ new parameters. To avoid any impact on the core model's performance, we train it as a second step where we freeze all parameters other than $\mathcal{M}$. We simply use a per-layer independent cross-entropy loss against a consistency oracle $\mathbb{1}[\arg\max(p(y_{t+1}|d_t^i) = \arg\max(p(y_{t+1}|d_t^L)]$, and average across the $L - 1$ layers. We also experimented with the geometric-like training of Elbayad et al. [23], but find it to be less effective here (see App. D). The two objectives are closely related, but the geometric one ignores any signal from the states post the first oracle exit.

## 4 Calibrating Local Early Exits from Global Constraints

We now describe our calibration procedure for finding a shared exit threshold $\lambda \in [0, 1]$ that can be used directly in Eq. (4), or via Eq. (5), such that we provably satisfy our desired global constraints over the fully generated sequences. At a high level, our approach uses the following basic recipe:

1. We specify a grid of possible values of $\Lambda = (\lambda_1, \ldots, \lambda_k)$ that *may* result in acceptable generations;
2. We choose the *lowest* valid $\lambda \in \Lambda$ that we can identify with rigorous statistical testing tools.

Let $P_{\text{test}}$ be an i.i.d. prompt given to the LLM at test time, and let $Y_{\text{full}} := \mathrm{LLM}_{\text{full}}(P_{\text{test}}) \in \mathcal{Y}$ and $Y_{\text{early}} := \mathrm{LLM}_{\text{early}}(P_{\text{test}}, \lambda) \in \mathcal{Y}$ denote the full and adaptive responses, respectively. Optionally, let $Z_{\text{test}}$ be a set of gold references for our task, if assumed. Our goal, as introduced in §1, is to find a valid $\lambda$ using $\mathcal{S}_{\text{cal}}$ such that we satisfy either of two types of global "consistency" constraints:

**Definition 1** (Textual consistency)**.** *An adaptive LLM is textually consistent if given any bounded text dissimilarity function,* $\mathcal{D} : \mathcal{Y} \times \mathcal{Y} \to \mathbb{R}$*, and tolerance* $\delta \in \mathbb{R}$*,* $\mathbb{E}\big[\mathcal{D}(Y_{\text{early}}, Y_{\text{full}})\big] \le \delta$*.*

**Definition 2** (Risk consistency)**.** *An adaptive LLM is risk consistent if given any bounded risk function,* $\mathcal{R} : \mathcal{Y} \times 2^{\mathcal{Y}} \to \mathbb{R}$*, and tolerance* $\delta \in \mathbb{R}$*,* $\mathbb{E}[\mathcal{R}(Y_{\text{early}}, Z_{\text{test}})] \le \mathbb{E}[\mathcal{R}(Y_{\text{full}}, Z_{\text{test}})] + \delta$*.*

Without loss of generality, we will assume that $\mathcal{D}$ and $\mathcal{R}$ are always normalized to the unit interval $[0, 1]$, and therefore will only be considering tolerances $\delta \in (0, 1)$. At a glance, to find a $\lambda$ that produces a consistent $\mathrm{LLM}_{\text{early}}$, we cast our problem as a multiple hypothesis testing problem over a large array of $k$ candidate classifier exit thresholds, $\Lambda = (\lambda_1, \ldots, \lambda_k)$, and apply the Learn then Test (LTT) framework of Angelopoulos et al. [3] to identify a subset of statistically valid, constraint-satisfying thresholds $\Lambda_{\text{valid}} \subseteq \Lambda$. Our final $\lambda$ is then chosen as $\lambda := \min(\Lambda_{\text{valid}} \cup \{1\})$.

### 4.1 The Learn then Test calibration framework

Choosing a value of $\lambda$ that rigorously satisfies our consistency objectives is challenging, as the performance impact of increasing or decreasing $\lambda$ is not necessarily monotonic. Naively setting $\lambda$, for example, based simply on average calibration set performance, can lead to statistically invalid results in our finite-sample, distribution-free setting. The LTT framework proposed by Angelopoulos et al. [3] solves this problem by reframing hyper-parameter selection as a multiple testing problem.

Let $\Lambda = (\lambda_1, \ldots, \lambda_k)$ be a finite grid of hyper-parameter values that may, or may not, obtain valid consistency. For example, when searching for a value of $\lambda \in [0,1]$, we might consider the evenly spaced set $\Lambda = \{\frac{i}{k+1} : i = 1, \ldots, k\}$. LTT then identifies a subset of values, $\Lambda_{\text{valid}} \subseteq \Lambda$, where

$$\mathbb{P}\Big(\exists \lambda \in \Lambda_{\text{valid}} : \text{LLM}_{\text{early}}(P_{\text{test}}, \lambda) \text{ and } \text{LLM}_{\text{full}}(P_{\text{test}}) \text{ are } \textbf{not} \text{ consistent}\Big) \leq \epsilon. \tag{7}$$

Here, we are using consistency to refer to either textual consistency or risk consistency. Eq. (7) can be satisfied by applying standard multiple hypothesis testing techniques as long as super-uniform p-values, $p_j$, are supplied for each value $\lambda_j \in \Lambda$ that support the null hypothesis

$$H_j : \text{LLM}_{\text{early}}(P_{\text{test}}, \lambda_j) \text{ and } \text{LLM}_{\text{full}}(P_{\text{test}}) \text{ are } \textbf{not} \text{ consistent.} \tag{8}$$

$\lambda_j$ is placed in $\Lambda_{\text{valid}}$ if $H_j$ is rejected, and discarded otherwise. This yields a consistent $\text{LLM}_{\text{early}}$.

**Proposition 1** (LTT for CALM). *Suppose $p_j$ is super-uniform for all $j$ under $H_j$ for some specified tolerance $\delta \in (0, 1)$. Let $\mathcal{A}$ be any family-wise error rate (FWER) controlling procedure at a level $\epsilon \in (0, 1)$, where $\mathcal{A}(p_1, \ldots, p_k)$ selects $H_j$ to reject. Choosing $\lambda := \min(\Lambda_{\text{valid}} \cup \{1\})$ then yields a consistent $\text{LLM}_{\text{early}}$ with probability at least $1 - \epsilon$.*

Note that a FWER-controlling procedure at a level $\epsilon$ is an algorithm that decides to accept or reject hypotheses $\{H_i\}_{i=1}^k$, while ensuring that the probability of falsely rejecting any $H_j$ is less than $\epsilon$. The proof of Proposition 1, given in Appendix A.1, follows directly from Theorem 1 of Angelopoulos et al. [3], and the fact that $\text{LLM}_{\text{early}}(P_{\text{test}}, 1) = \text{LLM}_{\text{full}}(P_{\text{test}})$ by construction per Eq. (4), so that we can always use $\lambda = 1$ as a valid fallback if we fail to identify non-empty $\Lambda_{\text{valid}}$. In the next sections, we describe how we calculate valid p-values using $\mathcal{S}_{\text{cal}}$, and our choice of FWER-controlling procedure.

### 4.2  Defining p-values for consistent early-exiting

LTT relies on valid p-values $p_j$, where $p_j$ is a random variable satisfying $\mathbb{P}(p_j \leq u) \leq u$ under $H_j$ for all $u \in [0, 1]$. For our purposes, we can obtain valid p-values from the empirical consistency of $\text{LLM}_{\text{early}}(P_i, \lambda)$ measured over the random calibration sample, $\mathcal{S}_{\text{cal}}$. Since we have assumed w.l.o.g. that either of our bounded consistency functions $\mathcal{D}$ and $\mathcal{R}$ from Defs. 1 and 2 have been normalized to lie in $[0, 1]$, we can, for example, obtain a valid p-value by simply inverting Hoeffding's inequality:[3]

$$p_j^{\text{Hoeffding}} := e^{-2n(\max(0, \delta - \widehat{E}(\lambda_j)))^2}, \tag{9}$$

where $\widehat{E}(\lambda_j) := \frac{1}{n} \sum_{i=1}^n L_i(\lambda_j)$ is the empirical average of random variable $L_i(\lambda_j) \in [0, 1]$, with

$$L_i(\lambda_j) := \mathcal{D}(\text{LLM}_{\text{early}}(P_i, \lambda_j), \text{LLM}_{\text{full}}(P_i)) \quad \text{or} \tag{10}$$

$$L_i(\lambda_j) := \max\left(0, \mathcal{R}(\text{LLM}_{\text{early}}(P_i, \lambda_j), Z_i) - \mathcal{R}(\text{LLM}_{\text{full}}(P_i), Z_i)\right), \tag{11}$$

for textual consistency versus risk consistency, respectively. Note that, as a technicality of enforcing the r.v. $L_i(\lambda_j)$ to be within $[0, 1]$, Eq. (11) computes a conservative estimate of the difference in the empirical risk that doesn't reward instances in which the risk of the early-exit model is lower.

### 4.3  Efficient fixed sequence testing

The more values of $\lambda$ we test, the higher the chance that we might accidentally choose a $\lambda$ that does not in fact result in consistent generations, despite whatever misleading performance we might have measured by chance on $\mathcal{S}_{\text{cal}}$. As part of LTT, we must select a multiple testing procedure that corrects for this (i.e., that controls the FWER at level $\epsilon$). Though the precise dependence between the early-exit LLM's performance and $\lambda$ is unknown, in practice we find that it tends to be fairly smooth and roughly monotonic. That is, nearby thresholds $\lambda \approx \lambda'$ tend to perform similarly, whereas $\lambda > \lambda'$ tends to result in relatively more consistent performance. Taking advantage of this structure, we choose to employ fixed sequence testing (FST) as our FWER-controlling procedure [3; 11].

Here we define a sequence of *descending* thresholds $\lambda_1 > \lambda_2 > \ldots \lambda_k$ with a relatively coarse step size (e.g., increments of 0.05). For each $\lambda_j$ *in order*, we compute $p_j$, and reject $H_j$ if $p_j \leq \epsilon$. The first time we fail to reject $H_j$, we immediately terminate our search, and return $\lambda_{j-1}$ to use as our calibrated threshold (or 1, if we fail to reject $H_1$). An Algorithm of the full procedure is provided in Appendix E.

---

[3]In practice, we use the more powerful Hoeffding-Bentkus bound [3; 10; 12; 33].

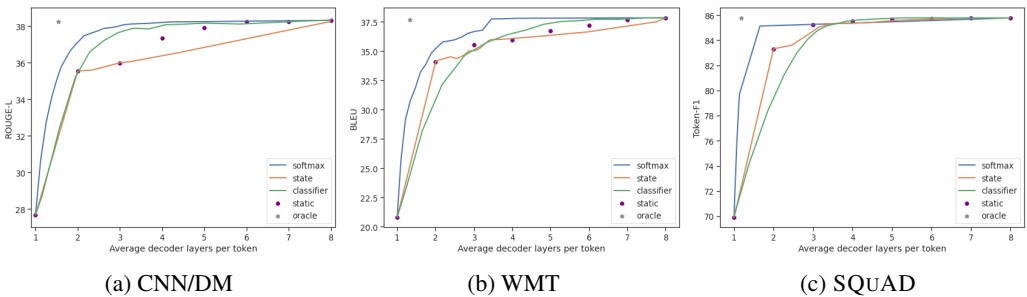

|  (a) CNN/DM | (b) WMT | (c) SQUAD |

Figure 3: Validation empirical performance-efficiency tradeoffs for different confidence measures, compared to static baselines and a local oracle measure with state propagation for skipped layers.

## 5   Experimental Setting

We empirically evaluate our methods on three popular text generation tasks that vary in their target generation length and extractive degrees against the input. **CNN/DM** [31] is a collection of news articles to be summarized in few sentences. **WMT15 EN-FR** [13] contains English sentences (one per example) to be machine translated to French. **Open-book SQUAD 1.1** [54] is a QA dataset with Wikipedia paragraphs paired with questions, where the target answer is a text span from the input. Length statistics of the validation sets are summarized in Table 1.

Table 1: Average number of tokens in reference targets of evaluation datasets (5/95th percentiles in parenthesis).

| Dataset | Output length |
| --- | --- |
| CNN/DM | 82 (42 - 141) |
| WMT EN-FR | 39 (10 - 82) |
| SQUAD | 5 (1 - 13) |

**Model.** We implement CALM on top of the T5 encoder-decoder model that showed good performance on the tasks above [53], using the T5X framework [55]. We use the 8 layers T5 1.1 model that doesn't share input and output embeddings. We share all output embeddings for the softmax predictions, and the early-exit classifier across all decoder layers. Based on validation results, we set the temperature of our decaying threshold to $\tau = 4$ for the softmax and classifier measures of CNN/DM and WMT. In other settings, we use $\tau = 0$. See App. C for more details, and App. B.3 for a 12 layers T5 model.

**Evaluation metrics.** We use the standard metrics for each task: ROUGE-L for CNN/DM, BLEU [50] for WMT, and Token-F1 [54] for SQUAD. We rely on the same metrics for computing the risk and textual distance, other than BLEU which is a corpus-level metric that doesn't directly enable expectation control. Instead, we use the BLEURT learned metric [61]. For a given metric $m(y_{\text{early}}, y_{\text{full}} \text{ or } z_{\text{test}}) \in [0, 1]$, we use $1 - m$ for distance or risk computation, respectively.

Our main efficiency metric is the average number of **decoder layers** used per output token, as it directly measures complexity reduction without conflating with implementation or infrastructure specific details [19]. For reference, we also report the average decoder **FLOPs reduction** per token [23]. Also, we compute an estimated **speedup** of the whole encoder-decoder model for generating the full sequence, based on TPUv3 benchmarking with 200 examples in Colab (see App. C for details).

**Calibration experiments.** For each task, we use the validation and test sets to evaluate our calibration method (§4) (for SQUAD we only use the validation set as the test answers are hidden). We run 50 random trials per target tolerance $\delta$ and consistency objective (textual or risk), where we partition the data to 80% calibration ($\mathcal{S}_{\text{cal}}$) and 20% test ($P_{\text{test}}$). We set $\epsilon = 0.05$ for all experiments.

**Baselines.** We emphasize that the CALM framework is general for any autoregressive multi-layered LM with any confidence measure, allowing controlled consistency by Eq. (1) or Eq. (2). To empirically evaluate the efficiency gains enabled by our proposed confidence measures, we compare with **static** baselines that use the same number of layers for all tokens. We also compare our early-exit classifier training with the geometric method of [23] in Appendix D. Also, we compute an **oracle** local measure (§3.3.1) as an upper-bound estimate of the performance-efficiency tradeoff.

## 6   Experimental Results

We first report the empirical performance-efficiency tradeoff achieved with each confidence measure. For each task and measure, we evaluate the full range of $\lambda$ on the validation set, with steps of 0.05.

Table 2: Test efficiency gains per choice of $\delta$, consistency objective, and confidence measure. $\epsilon$ is set to $0.05$. For plots of the full range of $\delta$ with standard deviation, see Appendix B.

| | $\delta$ | Measure | CNN/DM layers | CNN/DM FLOPs r. | CNN/DM speedup | WMT layers | WMT FLOPs r. | WMT speedup | SQUAD layers | SQUAD FLOPs r. | SQUAD speedup |
|---|---|---|---|---|---|---|---|---|---|---|---|
| Textual consist. | 0.1 | softmax | **5.73** | ×0.44 | ×1.41 | **3.35** | ×0.66 | ×2.01 | **1.65** | ×3.15 | ×1.63 |
| | | state | 8.00 | ×1.00 | ×1.00 | 7.68 | ×1.01 | ×1.00 | 2.00 | ×**3.65** | ×**1.68** |
| | | classifier | 7.16 | ×**1.03** | ×**1.42** | 5.50 | ×**1.06** | ×**2.05** | 2.59 | ×2.37 | ×1.10 |
| | 0.25 | softmax | **2.62** | ×0.49 | ×**2.57** | **1.76** | ×0.91 | ×**2.83** | **1.03** | ×**5.68** | ×**1.88** |
| | | state | 7.97 | ×1.00 | ×1.01 | 2.84 | ×**1.93** | ×1.55 | 2.00 | ×3.65 | ×1.68 |
| | | classifier | 4.51 | ×**1.15** | ×2.04 | 2.97 | ×1.22 | ×2.00 | 1.37 | ×5.09 | ×1.11 |
| Risk consist. | 0.02 | softmax | **3.75** | ×0.47 | ×**1.96** | **3.19** | ×0.67 | ×**2.10** | **1.65** | ×3.15 | ×1.63 |
| | | state | 7.97 | ×1.00 | ×1.01 | 7.68 | ×1.01 | ×1.00 | 3.13 | ×2.11 | ×**1.68** |
| | | classifier | 6.49 | ×**1.06** | ×1.71 | 5.05 | ×**1.08** | ×1.97 | 3.36 | ×1.55 | ×1.11 |
| | 0.05 | softmax | **1.73** | ×0.50 | ×**3.53** | **1.96** | ×0.85 | ×**2.73** | **1.65** | ×3.15 | ×1.63 |
| | | state | 5.22 | ×1.11 | ×1.64 | 2.72 | ×**2.01** | ×1.58 | 2.00 | ×**3.65** | ×**1.68** |
| | | classifier | 2.30 | ×**1.25** | ×2.09 | 3.08 | ×1.21 | ×1.98 | 2.59 | ×2.37 | ×1.10 |

The results, presented in Figure 3, show the power of the softmax response measure, allowing only minor performance loss while reducing more than half of the layers in all three tasks. The early-exit classifier, that is more FLOP-efficient, is also effective, mostly when targeting high performance (right hand side of plots). The simple and parameter-free state saturation measure is competitive, but often falls bellow the static baseline, despite enabling per-token exit decisions.

The dynamic oracle obtains compelling efficiency gains, using only 1.5, 1.3, and 1.2 layers on average for summarization, WMT, and QA, respectively, without losing any performance. This illustrates the full potential of CALM and leaves further room for improvements with better confidence measures. It also shows the effectiveness of inference-time state propagation for skipped layers (§3.3.1).

## 6.1 Calibrated performance with guaranteed textual or risk consistency

Next, we examine the outcomes of the calibration process. Since the obtained risk is guaranteed to be valid (i.e., $\leq \delta$ at least 95% of the time), we focus here on efficiency gains per chosen $\delta$. We refer the reader to Appendix B for empirical validation and for additional results and qualitative examples.

Table 2 presents the efficiency gains per choice of $\delta$ for each consistency objective and confidence measure. We examine larger $\delta$ values for textual consistency as this is generally a stricter requirement since the full model's error is not considered.

Across all, the softmax confidence measure leads to the greatest decrease in number of decoder layers required. Accordingly, softmax mostly enables the highest speedup gains of up to about three times faster than running through all the model's layers. The very lightweight early-exit classifier sometimes provides better gains than softmax, even if more decoding layers are used. Since the speedup is computed over the full generated output, we see more gains on the longer outputs of summarization and translation where the decoding takes most of the time, compared to the short QA outputs where the whole decoding time is not much longer than the encoding time.

These encouraging efficiency gains are enabled even with the rigorous performance guarantees that are sometimes conservative (e.g., Eq. (11)). We note that relaxing these constraints, or tightening the confidence intervals (e.g., with larger calibration sets), can further improve the empirical gains.

The softmax operation over the full output vocabulary is FLOPs heavy (though, this compute can potentially be paralleled), sometime leading to increased total FLOPs, even with fewer used layers. The state-based and early-exit classifier measures require minimal FLOPs and provide a good alternative with compelling efficiency gains, if total (parallelizable, or not) FLOPs is of concern.

## 6.2 Example output: effectively distributing the model's capacity across timesteps

Figure 4 presents two CALM summary generations for an article from the CNN/DM dataset, compared to the output of the full model (See Figure B.5 in the Appendix for examples from the other tasks). $Y_{\text{early}}^{(2)}$ uses a lower confidence threshold for early exiting compared to $Y_{\text{early}}^{(1)}$. The colors, depicting the number of decoder layers used per output token, illustrate how CALM obtains the

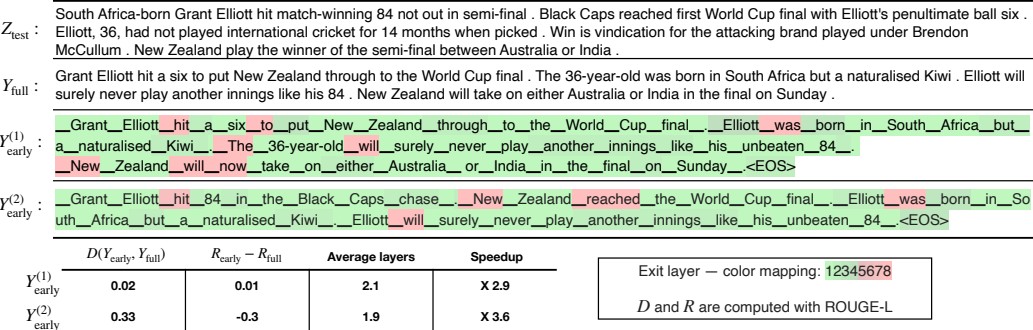

| | $Z_{\text{test}}$ : | South Africa-born Grant Elliott hit match-winning 84 not out in semi-final . Black Caps reached first World Cup final with Elliott's penultimate ball six . Elliott, 36, had not played international cricket for 14 months when picked . Win is vindication for the attacking brand played under Brendon McCullum . New Zealand play the winner of the semi-final between Australia or India . |
|---|---|---|
| | $Y_{\text{full}}$ : | Grant Elliott hit a six to put New Zealand through to the World Cup final . The 36-year-old was born in South Africa but a naturalised Kiwi . Elliott will surely never play another innings like his 84 . New Zealand will take on either Australia or India in the final on Sunday . |
| | $Y_{\text{early}}^{(1)}$ : | __Grant__Elliott__hit__a__six__to__put__New__Zealand__through__to__the__World__Cup__final__.__Elliott__was__born__in__South__Africa__but__ a__naturalised__Kiwi__.__The__36-year-old__will__surely__never__play__another__innings__like__his__unbeaten__84__. __New__Zealand__will__now__take__on__either__Australia__or__India__in__the__final__on__Sunday__.<EOS> |
| | $Y_{\text{early}}^{(2)}$ : | __Grant__Elliott__hit__84__in__the__Black__Caps__chase__.__New__Zealand__reached__the__World__Cup__final__.__Elliott__was__born__in__So uth__Africa__but__a__naturalised__Kiwi__.__Elliott__will__surely__never__play__another__innings__like__his__unbeaten__84__.<EOS> |

| | $D(Y_{\text{early}}, Y_{\text{full}})$ | $R_{\text{early}} - R_{\text{full}}$ | **Average layers** | **Speedup** |
|---|---|---|---|---|
| $Y_{\text{early}}^{(1)}$ | 0.02 | 0.01 | 2.1 | X 2.9 |
| $Y_{\text{early}}^{(2)}$ | 0.33 | -0.3 | 1.9 | X 3.6 |

Exit layer — color mapping: 12345678

$D$ and $R$ are computed with ROUGE-L

Figure 4: CALM accelerates the generation by early exiting when possible, and selectively using the full decoder's capacity only for few tokens, demonstrated here on a CNN/DM example with softmax-based confidence measure. $Y_{\text{early}}^{(1)}$ and $Y_{\text{early}}^{(2)}$ use different confidence thresholds for early exiting. Bellow the text, we report the measured textual and risk consistency of each of the two outputs, along with efficiency gains. The colors represent the number of decoding layers used for each token—light green shades indicate less than half of the total layers.

efficiency gains. Only a few selected tokens use the full capacity of the model (colored in red), while for most tokens the model exits after one or few decoding layers (colored in green).

The example in Figure 4 also demonstrates one difference between the two types of consistency constraints, given a reference output $Z_{\text{test}}$. Textual consistency $D(Y_{\text{early}}, Y_{\text{full}})$ generally (though, not always) degrades (i.e., increases) when decreasing the confidence threshold as the outputs tend to more significantly diverge from $Y_{\text{full}}$. The trend of risk consistency, however, depends also on the reference output $Z_{\text{test}}$. If $Y_{\text{full}} \approx Z_{\text{test}}$ then the two constraints are nearly the same. In this example, they are sufficiently different that $Y_{\text{early}}^{(2)}$ obtained better (lower) risk even though the textual distance from $Y_{\text{full}}$ is higher. On the one hand, given the availability of reference outputs for calibration, this suggests that for an imperfect model, risk consistency could lead to more aggressive early-exiting while maintaining the quality of generations. On the other hand, since the $Relu$ in Eq. (11) doesn't reward negative risk differences, the benefits might not fully materialize. Overall, the two constraints provide different alternatives for the user to choose from depending on the availability of reference outputs, the performance of the full model, and the exact desired performance guarantees.

# 7 Conclusion

We present confident adaptive language modeling (CALM) for dynamically allocating different amounts of compute per generated token, following explicitly defined tolerance levels on the full generation output. This paper covers both modeling solutions and analyses towards this goal, as well as a theoretically-grounded framework for provably controlling the quality of the full output to meet the user-specified tolerance levels. We investigate the effects of local early exiting during decoding on the final output, leading us to propose a decaying function over the initial threshold that enables finer control over the performance-efficiency tradeoffs without inflating the search space. We also study different solutions for addressing missing computations of early-exited tokens that are dependent upon for future tokens. Overall, our complete adaptive compute framework for LMs requires minimal modifications to the underlying model and enables efficiency gains while satisfying rigorous quality guarantees for the output. Also, our oracle experiments and runtime analysis demonstrates the full potential of this framework and leave room for future research to further improve the efficiency in a controllable way.

## Acknowledgements

We thank Ionel Gog for significantly improving the implementation after submission. We also thank Anselm Levskaya, Hyung Won Chung, Seungyeon Kim, Tao Wang, Paul Barham, and Michael Isard for great discussions and code suggestions. We thank Orhan Firat, Carlos Riquelme, Aditya Menon, Zhifeng Chen, Sanjiv Kumar, and Jeff Dean for helpful discussions and feedback on the project.

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
