# A  Mathematical Details

## A.1  Proof of Proposition 1

*Proof.* Define $\lambda$ to be risk-controlling if $\text{LLM}_{\text{early}}(P_{\text{test}}, \lambda)$ $\text{LLM}_{\text{full}}(P_{\text{test}})$ are consistent. Suppose $\Lambda_{\text{valid}}$ is non-empty. Then all $\lambda \in \Lambda_{\text{valid}}$ are risk-controlling w.p. $\geq 1-\epsilon$, per Thm. 1 of Angelopoulos et al. [3]. Furthermore, per Eq. (4) for $\lambda = 1$ we have $\text{LLM}_{\text{early}}(P_{\text{test}}, 1) = \text{LLM}_{\text{full}}(P_{\text{test}})$ by definition, since $\sup \mathcal{M}_i(h_i) = 1, \forall i \in [1, \ldots, L]$. Thus $\lambda = 1$ is also risk-controlling by definition. Combined, all $\lambda \in \Lambda_{\text{valid}} \cup \{1\}$ are risk-controlling w.p. $\geq 1-\epsilon$, and therefore $\lambda := \min(\Lambda_{\text{valid}} \cup \{1\})$ is always well-defined and guaranteed to be risk-controlling w.p. $\geq 1-\epsilon$. □

# B  Additional Results

We provide additional experimental results to supplement Section 6. In Section B.1, we include calibration plots, for both the validation and test sets, with the full range of $\delta$, also showing the standard deviation across random trials. In Section B.2, we present a few example outputs with a visualization of the per-token early-exit decisions to illustrate CALM's behavior. In Section B.3, we include results of a larger 12-layer model, showing the generalizability of our framework to other configurations.

## B.1  Calibration results for the full tolerance range

We present complementary results to Table 2. Figure B.1 and Figure B.2 present the empirical consistencies and efficiency gains for textual and risk consistency constraints, respectively. Figure B.3 and Figure B.4 report the same on the validation datasets. First, we observe that the calibration holds empirically, achieving risk values that are not greater than the specified $\delta$ (i.e., being under the diagonal in the upper subplots). We also see that the risk is often lower than allowed for (a good thing), especially with the risk consistency objective. This is due to the conservativeness of our measure (Eq. (11)), not rewarding instances where the early prediction has lower risk. While obtaining lower risk than the target is not a downside, this indicates that there is further potential in improving the efficiency gains achieved per $\delta$. Yet, even with the rigorous and conservative theoretical guarantees, we already obtain significant efficiency gains that, naturally, increase with larger tolerance values.

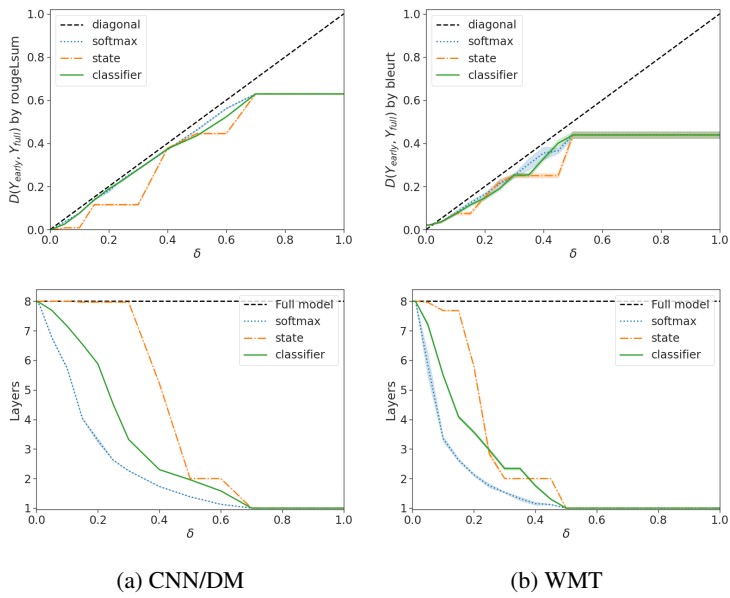

(a) CNN/DM                          (b) WMT

Figure B.1: Textual consistency and efficiency gains over the test sets per choice of $\delta$ ($\epsilon = 0.05$). The top row presents the empirical consistency where being under the diagonal means satisfying $\delta$ consistency. The bottom row presents the average number of decoder layer used. Shaded areas represent the standard deviation over random trials.

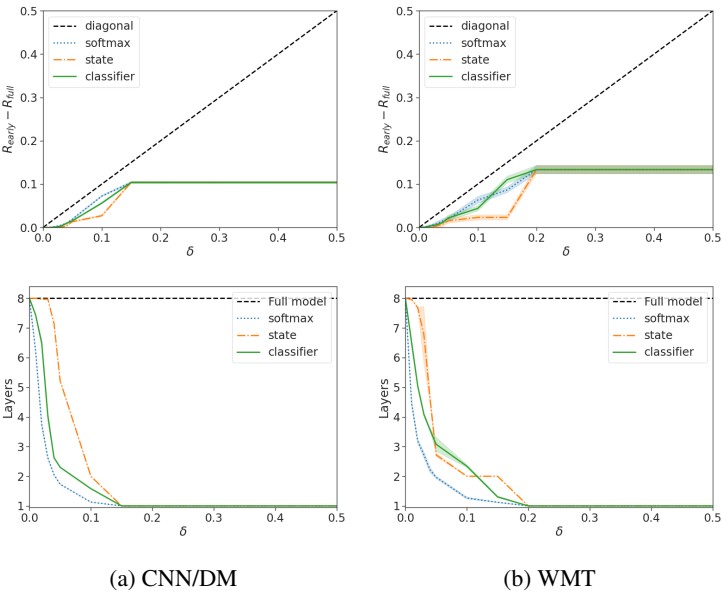

| (a) CNN/DM | (b) WMT |

Figure B.2: Risk consistency and efficiency gains over the test sets per choice of $\delta$ ($\epsilon = 0.05$). The top row presents the empirical consistency where being under the diagonal means satisfying $\delta$ consistency. The bottom row presents the average number of decoder layer used. Shaded areas represent the standard deviation over random trials.

## B.2  Qualitative examples

Figure B.5 presents two example outputs of CALM for instances from the machine translation, and question-answering (QA) datasets (See Figure 4 for summarization). The colors depict the per-token number of decoder layers that were used for generating that output. We also report the risk values for textual and risk consistency of both outputs, as well as the speedup compared to the full model. We observe that the textual distance generally increases as we accelerate the decoding. Though, the outputs still remain relatively similar to the full model even when using very few layers. The risk consistency doesn't always correlate with the textual one when the full model's risk is non-zero. In some cases, the accelerated output has even lower risk than the full model's output. This demonstrates the value of having both our textual and risk consistency configurations, which the user can pick from based on their objective, and whether quality reference outputs for calibration are available or not.

Interestingly, following our initial intuition, CALM distributes the compute unevenly, using very few layers for certain "easy" tokens, and additional compute to "hard" tokens. Examining the examples, we see that many times "hard" generation steps come at the beginning of sentences, or when generating a verb. We leave further investigations on perceived difficulties to future work.

## B.3  T5-base results

While throughout the rest of this paper we experiment with a 8-layer encoder-decoder T5 model. We include here results for a 12-layer T5-base model that besides the additional layers is also larger in its internal dimensions, having 12 attention heads and 64, 768, and 2048 dimensions for the attention head, embeddings, and MLP, respectively.

Figure B.6 shows the empirical performance-efficiency tradeoffs achieved with this model on the three tasks. Overall, we the trends are very similar to the one observed with the 8-layer model (Figure 3). One exception is the SQUAD model where the static baseline that uses only one or two decoder layers completely fails. This suggests that the actual predictions of this model are starting to be formed only from the third layer. Also, the local oracle measure on SQUAD obtains slightly lower global performance compared to the full model, also suggesting that in this case the hidden-state of the very low layers might not be a good enough representation for followup generations. Yet, the softmax and early-exit classifier confidence measure provide good proxies for the consistency

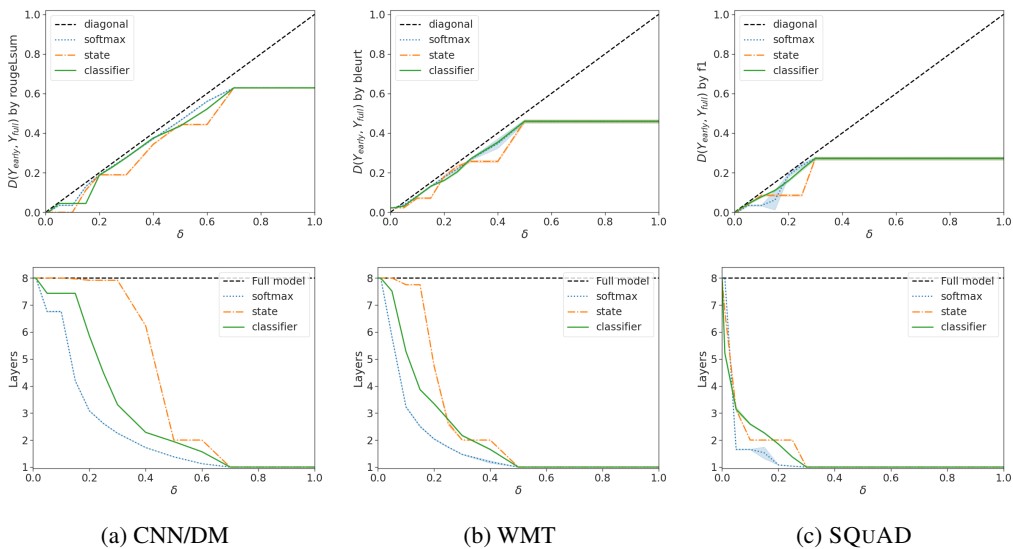

(a) CNN/DM         (b) WMT         (c) SQUAD

Figure B.3: Textual consistency and efficiency gains over the validation sets per choice of $\delta$ ($\epsilon = 0.05$). The top row presents the empirical consistency where being under the diagonal means satisfying $\delta$ consistency. The bottom row presents the average number of decoder layer used. Shaded areas represent the standard deviation over random trials.

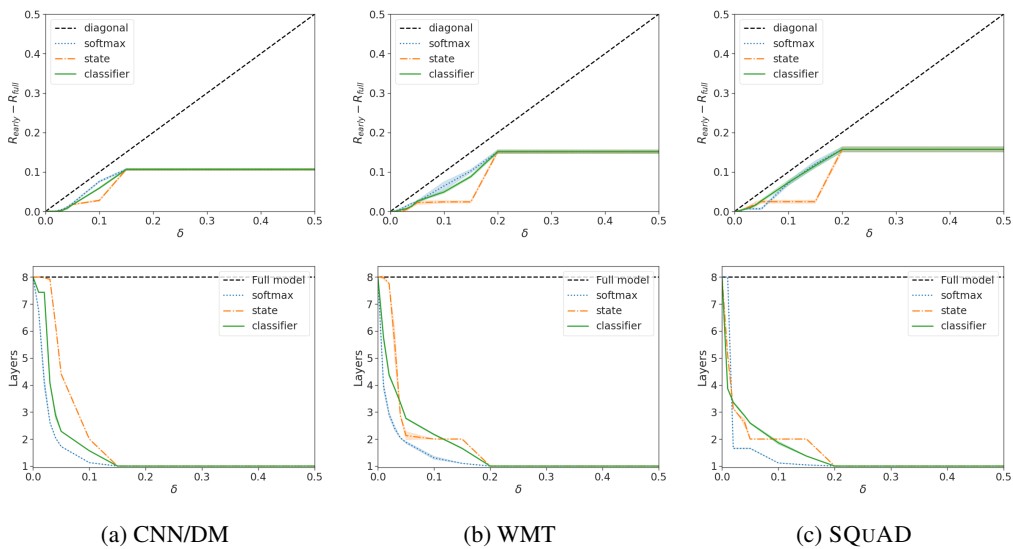

(a) CNN/DM         (b) WMT         (c) SQUAD

Figure B.4: Risk consistency and efficiency gains over the validation sets per choice of $\delta$ ($\epsilon = 0.05$). The top row presents the empirical consistency where being under the diagonal means satisfying $\delta$ consistency. The bottom row presents the average number of decoder layer used. Shaded areas represent the standard deviation over random trials.

and often outperform the static baselines. In the other two datasets, the local oracle matches the performance of the full model, similar to the behavior of the 8-layer model.

Figure B.7 and Figure B.8 present the validity and efficiency gains of our calibration procedure on the 12-layer model for textual and risk consistency objectives, respectively. We observe a largely similar behavior as the 8-layer model, showing the generality of our framework to other configurations of the backbone language model.

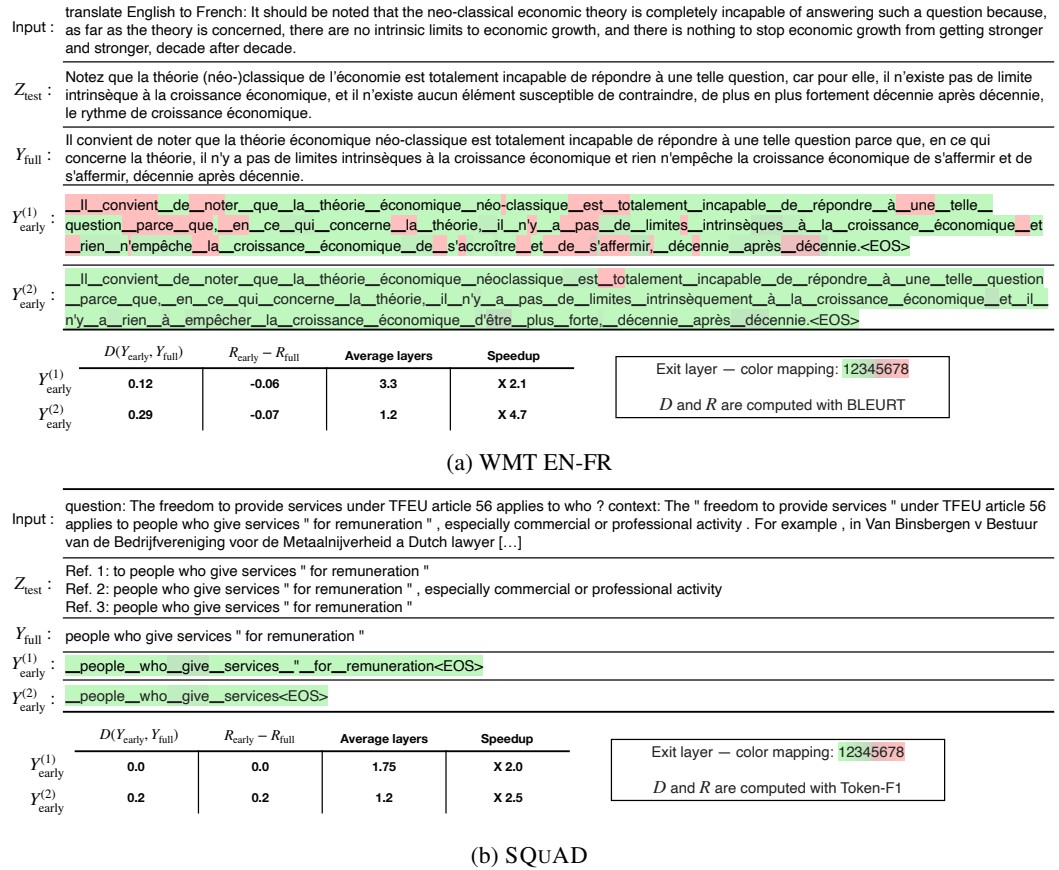

Input : translate English to French: It should be noted that the neo-classical economic theory is completely incapable of answering such a question because, as far as the theory is concerned, there are no intrinsic limits to economic growth, and there is nothing to stop economic growth from getting stronger and stronger, decade after decade.

$Z_{\text{test}}$ : Notez que la théorie (néo-)classique de l'économie est totalement incapable de répondre à une telle question, car pour elle, il n'existe pas de limite intrinsèque à la croissance économique, et il n'existe aucun élément susceptible de contraindre, de plus en plus fortement décennie après décennie, le rythme de croissance économique.

$Y_{\text{full}}$ : Il convient de noter que la théorie économique néo-classique est totalement incapable de répondre à une telle question parce que, en ce qui concerne la théorie, il n'y a pas de limites intrinsèques à la croissance économique et rien n'empêche la croissance économique de s'affirmer et de s'affirmer, décennie après décennie.

$Y_{\text{early}}^{(1)}$ : __Il__convient__de__noter__que__la__théorie__économique__néo-classique__est__totalement__incapable__de__répondre__à__une__telle__question__parce__que,__en__ce__qui__concerne__la__théorie,__il__n'y__a__pas__de__limites__intrinsèques__à__la__croissance__économique__et__rien__n'empêche__la__croissance__économique__de__s'accroître__et__de__s'affirmir,__décennie__après__décennie.<EOS>

$Y_{\text{early}}^{(2)}$ : __Il__convient__de__noter__que__la__théorie__économique__néoclassique__est__totalement__incapable__de__répondre__à__une__telle__question__parce__que,__en__ce__qui__concerne__la__théorie,__il__n'y__a__pas__de__limites__intrinsèquement__à__la__croissance__économique__et__il__n'y__a__rien__à__empêcher__la__croissance__économique__d'être__plus__forte,__décennie__après__décennie.<EOS>

| | $D(Y_{\text{early}}, Y_{\text{full}})$ | $R_{\text{early}} - R_{\text{full}}$ | Average layers | Speedup |
|---|---|---|---|---|
| $Y_{\text{early}}^{(1)}$ | 0.12 | -0.06 | 3.3 | X 2.1 |
| $Y_{\text{early}}^{(2)}$ | 0.29 | -0.07 | 1.2 | X 4.7 |

Exit layer — color mapping: 12345678

$D$ and $R$ are computed with BLEURT

(a) WMT EN-FR

Input : question: The freedom to provide services under TFEU article 56 applies to who ? context: The " freedom to provide services " under TFEU article 56 applies to people who give services " for remuneration " , especially commercial or professional activity . For example , in Van Binsbergen v Bestuur van de Bedrijfvereniging voor de Metalnijverheid a Dutch lawyer […]

$Z_{\text{test}}$ : Ref. 1: to people who give services " for remuneration "
Ref. 2: people who give services " for remuneration " , especially commercial or professional activity
Ref. 3: people who give services " for remuneration "

$Y_{\text{full}}$ : people who give services " for remuneration "

$Y_{\text{early}}^{(1)}$ : __people__who__give__services__"__for__remuneration<EOS>

$Y_{\text{early}}^{(2)}$ : __people__who__give__services<EOS>

| | $D(Y_{\text{early}}, Y_{\text{full}})$ | $R_{\text{early}} - R_{\text{full}}$ | Average layers | Speedup |
|---|---|---|---|---|
| $Y_{\text{early}}^{(1)}$ | 0.0 | 0.0 | 1.75 | X 2.0 |
| $Y_{\text{early}}^{(2)}$ | 0.2 | 0.2 | 1.2 | X 2.5 |

Exit layer — color mapping: 12345678

$D$ and $R$ are computed with Token-F1

(b) SQUAD

Figure B.5: Example outputs of CALM using a softmax-based confidence measure. Bellow the text, we report the measured textual and risk consistency of each of the two outputs, along with efficiency gains. See Figure 4 for an example from the CNN/DM task.

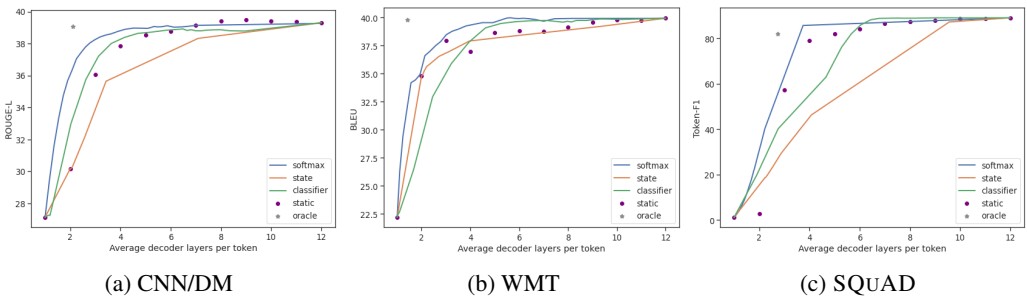

(a) CNN/DM          (b) WMT          (c) SQUAD

Figure B.6: T5-base (12 layers) validation empirical performance-efficiency tradeoffs for different confidence measures, compared to static baselines and a local oracle measure.

# C    Implementation Details

As mentioned in Section 5, we build on the T5 encoder-decoder model [53], and us the T5X repository [55] for implementing CALM. Appendix E describes the main algorithmic components.

Our main experiments use the T5 1.1 version with 8 layers for both the encoder and decoder modules, 6 attention heads with dimensions of 64, 512, and 1024 for the attention head, embeddings, and MLP, respectively. The vocabulary contains 32,128 tokens. This model doesn't share the input and output embeddings. For our early-exit head, we share the output embeddings between all intermediate with the top one, not introducing any new parameters to the model. Our binary early-exit classifier is also

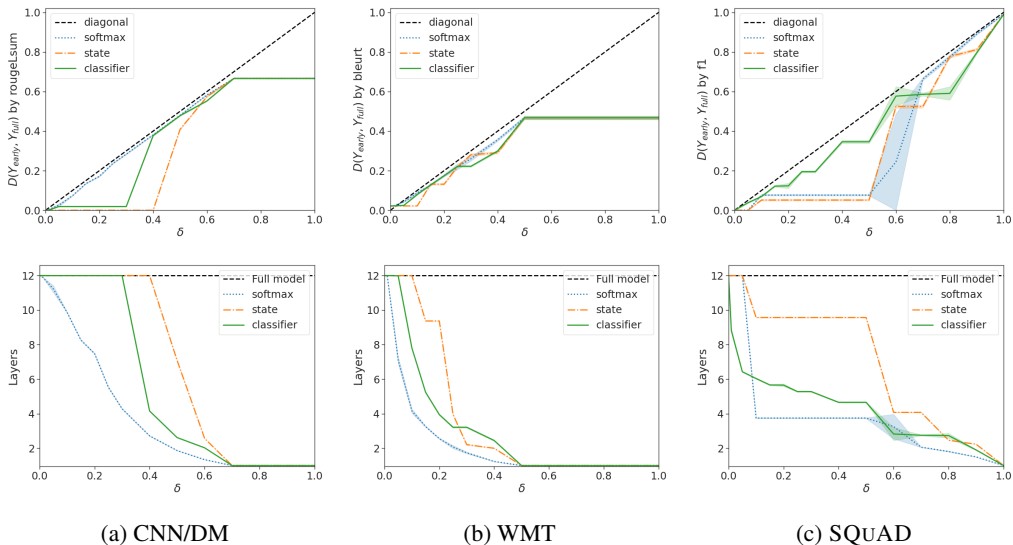

(a) CNN/DM      (b) WMT      (c) SQUAD

Figure B.7: Textual consistency and efficiency gains with a 12-layer T5-base model over the validation sets per choice of $\delta$ ($\epsilon = 0.05$). The top row presents the empirical consistency where being under the diagonal means satisfying $\delta$ consistency. The bottom row presents the average number of decoder layer used. Shaded areas represent the standard deviation over random trials.

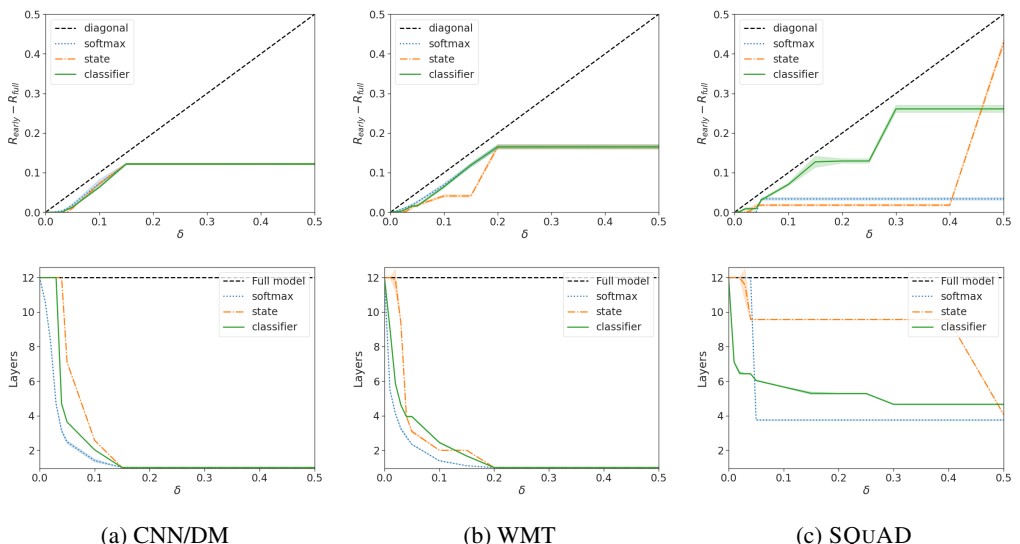

(a) CNN/DM      (b) WMT      (c) SQUAD

Figure B.8: Risk consistency and efficiency gains with a 12-layer T5-base model over the validation sets per choice of $\delta$ ($\epsilon = 0.05$). The top row presents the empirical consistency where being under the diagonal means satisfying $\delta$ consistency. The bottom row presents the average number of decoder layer used. Shaded areas represent the standard deviation over random trials.

shared across all layers, adding only a very small amount of new parameters. We add early-exit heads to all layers.

We fine-tune the models on the training set of each task for a maximum of 500K steps, and choose the best checkpoint by performance on the validation set (using the full models' predictions). We use a batch size of 128, the regular LM cross-entropy loss, the AdaFactor optimizer [64], and experiment with learning rates $10^{-3}$ and $10^{-4}$. We aggregate the loss of individual layers with a weighted average, as discussed in Section 3.4. For the early-exit classifier training, we use an unweighted average (See Appendix D for more details). We use 64 TPUv3 chips for training. For inference,

we use a single TPUv4 chip with a batch size of one, simulating a one-at-a-time processing setting, that is convenient when serving models for online requests. As described in Section 3.2, CALM exits early whenever the per-token confidence value $c$ (Section 3.5) exceeds the possibly-decaying threshold $\lambda$ (Section 3.3.2) derived from the user-defined $\delta, \epsilon$ tolerance levels and textual or risk consistency objective (Section 4). If necessary, the hidden-state is propagated to the skipped layers (Section 3.3.1).

## C.1 FLOPs and speedup computations

We detail our procedures for approximating the reference efficiency gains using the FLOPs and speedup measures. For FLOPs computation, to be consistent with Elbayad et al. [23] (See their Appendix B), we adopt their formula to compute the average decoder FLOPs per output token.

To measure the speedup of early exiting with CALM, we execute 200 inference predictions for each task under each examined configuration in a JIT compiled function in colab with TPUv3. We ignore the time of the first execution, since it is drastically slower due to compilation, and average the rest of the measured times. For each inference prediction, we use batch size one, and measure the full generation time including both the encoder and all decoding steps until completion. For each examined confidence measure (softmax, state, or classifier), we compute the speedup by comparing to the average inference time of the full model that uses all layers, by setting the confidence threshold to maximum. We note that our implementation adds conditionals between layers that add some overhead to the compute graph. Yet, even compared to a conditional-free implementation, the gains of CALM's early exits often outweigh the overheads of the added conditionals. We leave studying further technical improvements to the implementation to future work.

## D   Training Details of the Early Exit Classifier

Table D.1: $F_1$ scores of early-exit classifier for predicting not to exit. Measured at layers 1,4, and 7.

| Training method | CNN/DM | | | WMT | | | SQuAD | | |
|---|---|---|---|---|---|---|---|---|---|
| | $i = 1$ | 4 | 7 | $i = 1$ | 4 | 7 | $i = 1$ | 4 | 7 |
| Geometric-like | .59 | .35 | .24 | .33 | .18 | .11 | .73 | .16 | .11 |
| Independent | .62 | .49 | .37 | .51 | .34 | .24 | .76 | .26 | .18 |

As discussed in Section 3.5, we train the early exit classifier to predict whether the top-ranked prediction of a specific intermediate layer is the same as the prediction of the top layer. This classifier is using the intermediate hidden-state as its input features. The target labels are computed on-the-fly following an oracle that compares the intermediate prediction head with the prediction of the full model ($\mathbb{1}[\arg\max(p(y_{t+1}|d_t^i) = \arg\max(p(y_{t+1}|d_t^L)])$). We use the same training hyper-parameters used for the full model, but freeze all parameters of the backbone model (to eliminate any affect on the model's performance) and only train the newly added parameters for early-exit classifier, which are shared across all layers.

Following the setup above, we compute the binary cross-entropy loss for each layer individually, and aggregate by taking the unweighted average across all layers. We use the loss value on the validation set to pick the best checkpoint. We also explore with the geometric-like objective proposed by Elbayad et al. [23]. Their approach views the exiting decisions as a Bernoulli process, using the "exit"/ "don't exit" predicted probabilities. The goal is to make an "exit" prediction at the first true layer, determined by the oracle, and "don't exit" predictions by all preceding layers. Accordingly, the training objective maximizes the probability of this oracle-guided event, modeled as a product of all respective predicted probabilities. In practice, due to numerical instability of this product operation, we maximize the summation over the log probabilities.

Table D.1 presents the $F_1$ validation scores of "don't exit" predictions (with a 0.5 threshold) by early-exit classifier, measured against the oracle for layers 1,4, and 7. Our per-layer independent training objective outperforms the geometric-like objective across all layers and tasks. The advantage is typically most pronounced for higher layers. We conjecture that this is due to the equal weight of the independent objective that utilizes signal from all layers, whereas the geometric-like objective only learns from layers up to the first oracle exit.

# E Algorithms and Code

---

**Algorithm 1** Calibrating CALM for global consistency within $\delta, \epsilon$ tolerance levels with FST-based LTT [3].

---

1: **function** CALIBRATE($\text{LLM}_{\text{early}}, \text{LLM}_{\text{full}}, \delta, \epsilon$)
2:      $\lambda_{\min} = 1$
3:      $\Lambda = (\lambda_1, \lambda_2, \ldots, \lambda_k)$            ▷ Arrange decreasing candidate thresholds, $\lambda_i > \lambda_j \ \forall i < j$.
4:      **for** $\lambda_j \in \Lambda$ **do**
5:          $\widehat{E}(\lambda_j) = \frac{1}{n} \sum_{i=1}^{n} L_i(\lambda_j)$       ▷ Following Eqs. (10) or (11) for textual vs. risk consistency.
6:          $p_j = \exp(-2n(\max(0, \delta - \widehat{E}(\lambda_j)))^2)$ ▷ Compute p-value. Can replace with Hoeffding-Bentkus.
7:          **if** $p_j > \epsilon$ **then**
8:              **return** $\lambda_{\min}$
9:          $\lambda_{\min} = \lambda_j$
10:     **return** $\lambda_{\min}$

---

Algorithm 1 describes the calibration process of CALM for obtaining global textual or risk $\delta, \epsilon$ consistency (Section 4).

The JAX [14] code for training CALM models and for executing the early-exit functionality at inference-time is available at: `https://github.com/google-research/t5x/tree/main/t5x/contrib/calm`