# OpenReview forum: "Confident Adaptive Language Modeling"
_NeurIPS.cc/2022/Conference — NeurIPS 2022 Accept_

### Official Review · Reviewer_Jyd3 · 2022-07-05

**Rating:** 7
**Confidence:** 3
**Soundness:** 4 excellent
**Presentation:** 4 excellent
**Contribution:** 3 good

**Summary:**

**The research problem:** The authors study the problem of accelerating transformer-based autoregressive inference from the perspective of early-exit strategies - approaches that terminate the forward pass before the final layer is reached based on whether a confidence criterion is satisfied. The primary goal of the proposed framework, CALM, is to find a method that allows for exiting the forward passes as early as possible while guaranteeing performance bounds on the consistency between the full model and early-exit model.

**Analysis and proposed method:** After analyzing the sources of error that arise when running early-exit models, the authors propose a decaying confidence threshold for early exit that allow for earlier exits closer to the end of sequences.

**Theoretical guarantees on performance:** The authors use tools from multiple hypothesis testing literature to pick confidence thresholds that provably satisfy performance bounds.

**Empirical Justification:** The proposed method is shown to (with some caveats) result in potential speedup of up to 3x.

**POST REBUTTAL UPDATE:** I thank the authors for their responses. I've raised my score.


**Questions:**

* Could you elaborate on the meaning of the result reported on line 166? Am I right in thinking that if one uses only the first layer representations to attend to during decoding (as if all the previous tokens exited after the first layer), the performance basically doesn't take a hit? This seems highly counter-intuitive?
* Does it ever happen that the early-exit prediction matches that of the full model at layer i < L, then starts disagreeing again at layer j where  i < j < L?
* Line 201: "Perfect local consistency implies global consistency". Could you clarify what is meant here?



Nitpick:
* $\mathcal{Y}$ doesn't seem to be defined anywhere.
* Line 194: "Our goal" instead of "are goal"

**Limitations:**

The authors are upfront about the computational cost of using the softmax-based confidence measure (the best performing metric explored in the paper), which could potentially surpass the gains of the proposed method.



**Strengths And Weaknesses:**

**STRENGTHS**
* Provable performance bounds is appealing: The distribution free approach to proving performance bounds is appealing. The fact that this approach doesn't yield too-conservative bounds is promising.
* Approach considers local and global performance metrics: The proposed method considers the deterioration of quality both locally and globally when one employs early-exit strategies.
* Error analysis provides insight about transformer inner-workings: Experiments in Section 3.3 are quite interesting - that one can get away with using only 1.53 layers per forward pass on average in a 8-layer transformer while (almost) not incurring a significant performance deterioration is surprising.
* No labelled data is needed: The proposed method, when using textual consistency, doesn't require any labelled data.
* Comparison of different confidence measures provides insight into early-exit strategies: The comparisons between softmax, state and classifier based early-exit confidence measures is quite interesting
* Clear writing: The paper is easy to follow and well written, with very few typos.


**WEAKNESSES**
* Comparison to baselines: While the contribution of this work is to facilitate early-exit while providing performance guarantees, it'd still be interesting to see comparisons with baselines to see how much (if any) cost one has to pay to obtain the provable guarantees.
* Computational cost of some confidence measures: The best performing confidence measure (softmax-based one) is computationally heavy, potentially erasing the efficiency gains of early exit. (The authors do note that further parallelization can alleviate this issue, and other confidence measures also bring nontrivial improvements in efficiency)
* Practical benefit of performance guarantees in early-exit strategies: I believe the authors could spend some time justifying why one would be interested in provable guarantees on early-exit performance deterioration. What are some applications where this type of guarantee is desired/required? We already don't expect to have performance guarantees on the full model - what's the marginal benefit of using an early exit strategy that have theoretical guarantees instead of simply doing empirical validation to pick $\lambda$? This will probably be especially true in cases where full model (i.e. model without early-exit) performance is sub-optimal which will be the case for many difficult NLP tasks with even today's strongest models). (It's possible I'm missing something here).

---

> ### Author Response · Authors · 2022-08-01
> **Thank you**
>
> We thank the reviewer for the helpful comments. We respond to your questions below:
>
> > *Comparison to baselines*
>
> In Figure 3 we compare to static baselines that use the same number of layers for all tokens. We also report the local oracle baseline as an approximate upper bound. We note that most previous papers on early exiting develop methods for encoder-only (BERT-style) models. Therefore, their methods aren’t applicable to the autoregressive language models that we focus on here. Elbayad et al. is the most closely related work (they experimented on machine translation) and we extensively discuss it along the paper. Unfortunately, their code and models aren’t available so we couldn’t directly compare with their model. Yet, we reimplemented their best model (Tok-C Geometric-like) and found our independent training of the early exit classifier to outperform it (We report the results in Appendix D). We also share code snippets in the Appendix and plan to share the full code and models to facilitate future research and comparisons.
>
> > *Practical benefit of performance guarantees in early-exit strategies [...]*
>
> As also replied to Reviewer c1DX (who shared the same question), quality control can be a main blocker for adopting efficient solutions in practice. Users want to deploy good models; even small discrepancies in performance can amount to large material differences at scale. We believe that rigorous statistical guarantees—that are distribution-free, model-free and hold in finite-samples—are not just “nice to have,” but rather critical for building trust in (sometimes finicky and hard-to-understand) dynamic, amortized models. Our hope is that these ideas can help lead to wider adoption of efficient, yet performant (programmable via any tolerance the user wishes to set!), models.
>
>
> > *Could you elaborate on the meaning of the result reported on line 166? Am I right in thinking that [...]*
>
> Yes. It is important to note that for this to work properly, each layer has to use its own projection matrices K, V to project d for the self-attention module. As we mention in the following paragraph in the paper, copying the post-projection states from the exited layer leads to a significant drop in performance. Also, our multi-layered objective (Equation 6) might have an effect on the model’s behavior. Indeed, this is a very encouraging finding that enables the efficient teacher forcing training, while applying the early exiting strategy at inference time and effectively skipping unnecessary computations.
>
> > *Does it ever happen that the early-exit prediction matches that of the full model at layer i < L, then starts disagreeing again at layer j where i < j < L?*
>
> We see that most of the time (about 93% of predictions on the CNN/DM dataset), the prediction saturates and doesn’t change after the first match with the full model. In the few cases where it doesn’t, typically the prediction changes to something else at the next layer and then back to the full model’s prediction and saturates until the end.
>
> > *Line 201: "Perfect local consistency implies global consistency". Could you clarify what is meant here?*
>
> By perfect local consistency, we mean that the early-exit model outputs every single token in the sequence exactly like the full model would have. Since in all common sequence-level measures (e.g., Rouge, token-F1, etc.) two identical sequences always receive a perfect matching score (i.e., zero distance), outputting an identical sequence would mean that the two outputs are globally consistent. Note that the other direction doesn’t necessarily apply. For example, an early-exit model that outputs a sentence that has flipped the order of two adjectives will have zero token-F1 distance from the respective output of the full model (that only has the adjectives in the reversed order). Yet, the local prediction of the first adjective in the sentence will be locally inconsistent between the two models since the predicted tokens are different.
>
> > *Y doesn't seem to be defined anywhere.*
>
> Thank you for noticing. We will clarify that this is the space of possible output sequences up to a predefined maximum length.
>
> > *Line 194: "Our goal" instead of "are goal"*
>
> Thank you for noticing. Will correct the typo.

---

### Official Review · Reviewer_FJ71 · 2022-07-10

**Rating:** 8
**Confidence:** 4
**Soundness:** 4 excellent
**Presentation:** 3 good
**Contribution:** 3 good

**Summary:**

UPDATE: The authors have answered my concerned so I increased my score to an 8.

Transformer language models such as T5 are the most important tool in the current NLP world. Typically the decoder has a set number of layers and all of these layers are executed for each token. In the past few years we've seen many papers that do 'early exit' for tokens for which the output is easy to compute- during these timesteps, the model exists early and does not compute all the layers.

This paper first analyzes these types of model with an oracle to show what the best case speedups could be, and then proposes a few new ways to do early exit, for the T5 model, evaluated on a group of interesting downstream tasks such as summarization and translation.

I felt like the paper wasn't very clear in comparing the final performance to the many prior works such as Elbayad et al or Schwartz et al. In fact Table 2 which is the main table that presents results only contains the 3 proposed models, it does not present the full model (no early exit) baseline and it does not present prior models such as Elbayad et al or Schwartz et al.

In addition, while these models have been proposed for the past couple of years, none of them have been adopted or used in production. I feel like this might be because of many issues with these methods that none of these papers really discuss. For example, you point out the "potential FLOP speedup", but what is the *actual* wall clock speedups of your methods? When doing batched inference, these methods can't really be used because every token in the batch might require a different amount of compute. It would be good to discuss these potential issues in the next version of the paper.

**Questions:**

My questions are in the prior sections. The most important question is how would table 2 look if you added numbers from the full (no early-exit) model baseline and from the Elbayad and Schwartz papers.

**Limitations:**

The authors adequately addressed the limitations and potential negative societal impact of their work.

**Strengths And Weaknesses:**

Strengths:
1) The analysis in this paper is super interesting, the oracle analysis shows that these methods might have big potential.

Weaknesses:
1) No wallclock time. Sure- you might save some compute by not computing some layers. But you are also using more compute to decide when to stop running the layers. So what is the total running time? Providing wallclock times is the only way to show if these methods are actually efficient or not.
2) Can this method be used in batched inference? Is there any downside to that?
3) Table 2 should have comparisons with the full (no early-exit) model baseline and with models from previous papers. It's not clear where you stand in comparison to the prior literature.

---

> ### Author Response · Authors · 2022-08-01
> **Thank you**
>
> We thank the reviewer for the helpful comments. We respond to your questions below:
>
> > *No wallclock time.*
>
> The wallclock time will depend on the type of infrastructure and hardware used. For this reason, we use the average number of decoder layers used as our main efficiency metric. We also report the total reduction in decoder FLOPs in Table 2, while also accounting for any FLOPs accrued by our early-exit classification mechanism (e.g., the softmax response). By focusing on algorithmic solutions, we hope that this work will further motivate and facilitate infrastructure and hardware developments for adaptive compute solutions (e.g., as they have done for sparse matrix operations).
>
> That said, per request, we have measured the wallclock time of our current implementation. We benchmarked with a TPUv4 to get the average end-to-end inference time of a compiled model that uses the early exit classifier (third rows in Table 2). For CNN/DM the full model took 83.2 ms on average. CALM wallclock time was 75.5 ms and 55.8 ms for textual consistency with $\delta=0.1$ and $\delta=0.25$, respectively, and 69.9 ms and 45.3 ms for risk consistency with $\delta=0.02$ and $\delta=0.05$, respectively. For SQUAD, where the output sequences are much shorter, inference with the full model took 7.1 ms on average. CALM wallclock time was 3.7 ms and 3.3 ms for textual consistency with $\delta=0.1$ and $\delta=0.25$, respectively, and 4.2 ms and 3.7 ms for risk consistency with $\delta=0.02$ and $\delta=0.05$, respectively. Note that this implementation doesn’t parallelize the state propagation which can yield further speedup, especially when the exit is at a low layer.
>
> > *Can this method be used in batched inference?*
>
> Following previous papers on early-exiting (e.g., [Schwartz et al., 2020](https://arxiv.org/abs/2004.07453); [Elbayad et al., 2020](https://arxiv.org/abs/1910.10073)), our experiments in the paper are with batch size one. This is a realistic scenario when serving generative models since many applications have to serve one-at-a-time prompts from the user. Especially when latency is of concern (which is our target use case), waiting to aggregate queries into a batch can cause undesired delays.
>
> That said, our method can be trivially extended to larger batches by taking either the average or the minimum (to be conservative) confidence across the batch. This might reduce the per-example efficiency gains, but can in return benefit from the gains of using batches when possible. Another option is to dynamically reduce the batch size across the network as per-instance predictions reach an early exit decision. Finally, very recent developments in the accelerator literature like [Barham et al., 2022](​https://arxiv.org/abs/2203.12533) open the door to new capabilities that might allow dynamic routing per example within a batch. We leave this exploration to future work.
>
>
> > *Table 2 should have comparisons with the full (no early-exit) model baseline and with models from previous papers.*
>
> We emphasize that Table 2 presents results with respect to the full model. The reported FLOPs reduction and speedup factors are against the full (no early-exit) model. Furthermore, the exact performance of the full model, as well as other static baselines with fewer layers, is presented in Figure 3. The rightmost purple dot in the plot is the full no early-exit model.
>
> Regarding comparison to Elbayad et al or Schwartz et al., we extensively describe the relation to these papers in the related work section. Schwartz et al., along with many other recent papers, develop methods for encoder-only (BERT-style) models. Therefore, their methods aren’t applicable to the autoregressive language models that we focus on here. Elbayad et al. is indeed the most closely related work (they experimented on machine translation) and we extensively discuss it along the paper. Unfortunately, their code and models aren’t available so we couldn’t directly compare with their model. Yet, we reimplemented their best model (Tok-C Geometric-like) and found our independent training of the early exit classifier to outperform it (We report the results in Appendix D). We also share code snippets in the Appendix and plan to share the full code and models to facilitate future research and comparisons.

---

> > ### Author Response · Authors · 2022-08-01
> > **Continued**
> >
> > > *while these models have been proposed for the past couple of years, none of them have been adopted or used in production.*
> >
> > First, we note that the vast majority of previous models were encoder-only classifiers. Here, we focus on autoregressive text generation where the decoding part adds an extensive compute overhead due to the lon
> > g sequence of conditional computations. Second, over the past few years, we have seen an unprecedented and exponential growth in the size of Language Models—yet there are many challenges in using such large models in production. The goal of our work is to preserve the high performance of large models, while reducing the average cost. Finally, a core contribution of our work is to take a principled approach to the entire process so that it is performed reliably, with limited performance degradation. We believe that early-exiting, together with other algorithmic and infrastructure improvements, can be a key component in enabling the production of larger language models in the future.

---

### Official Review · Reviewer_c1DX · 2022-07-10

**Rating:** 7
**Confidence:** 3
**Soundness:** 4 excellent
**Presentation:** 3 good
**Contribution:** 3 good

**Summary:**

This paper introduces the "Confident Adaptive Language Modeling" (CALM) framework, which contains very detailed study about the source of errors of doing early exiting during the decoding process of Transformer based language model, and proposes a method to calibrate well about local and global predictions to overcome these errors. Several empirical results, on text summarization, machine translation and Squad demonstrated the usefulness of the proposed method.



**Questions:**

1. Would adding more softmax operators for each layer, for example listed in Equation (6), trigger more computational bottleneck? Understand that seemingly the trade-off here would be good since we can ignore a lot of full computations for the subsequent layers if we can early exit, but still Eqn (6) seems introducing the extra softmax operators for all the layers and all the timesteps, during training, and it would be good if there could be some ablation study here demonstrating the ROI;

2. It is definitely technically sound that the LTT framework is used. But this seems to be a double-sided sword: to determine the \lambda value in Eqn (5), is it really necessary to adopt such a complicated framework? Note supposedly doing complicated tech works might hurt the generalization of the proposed method in the real world application scenario. What if this problem is simply treated as a hyper-parameter selection one and use much simpler method to do so (e.g., some random search and early stopping methods used in AutoML)?

**Ethics Review Area:**

["I don’t know"]

**Limitations:**

See above. N/A for the "negative societal impact of their work" part.

**Strengths And Weaknesses:**

Strengths:

- This paper is technically very rigorous. It tackles the problem in a step-by-step manner with clear very sound logic chain. Also, it is not a common practice now to pay attention to the statistical testing criteria and this paper does a good job here, by adopting a principled way to address that.  The writing, including the mathematical notation system, is also very accurate.

- The technical works, especially in section 3.3,  are very detailed and (hence) insightful. For example, the headroom analysis of state propagation and local errors are comprehensive and convincing.

Weakness:

- See the below question 2 in the "Questions" part: it seems good to discuss whether the technical proposal in section 4, to solve this problem, is really necessary to be so complicated.

---

> ### Author Response · Authors · 2022-08-01
> **Thank you**
>
> We thank the reviewer for the helpful comments. We respond to your questions below:
>
> > *Would adding more softmax operators for each layer, for example listed in Equation (6), trigger more computational bottleneck? [...]*
>
> Adding more softmax layers can be performed quite efficiently. During training time, rather than executing L softmax calls sequentially, we simply collect all pre-softmax hidden states from all intermediate layers into one matrix, and then compute the final softmax operation for all layers in parallel. During inference time, however, we must compute softmaxes sequentially. The reviewer is correct that this adds significant computation in itself; we account for this in our FLOPs results, this is also why we propose other confidence measures (such as the early exit classifier) that have less overhead. That said, its better confidence signal can still allow us to save substantial time by exiting early.
>
> > *[...] to determine the \lambda value in Eqn (5), is it really necessary to adopt such a complicated framework? Note supposedly doing complicated tech works might hurt the generalization of the proposed method in the real world application scenario. [...]*
>
> Quality control can be a main blocker for adopting efficient solutions in practice. Users want to deploy good models; even small discrepancies in performance can amount to large material differences at scale. We believe that rigorous statistical guarantees—that are distribution-free, model-free and hold in finite-samples—are not just “nice to have,” but rather critical for building trust in (sometimes finicky and hard-to-understand) dynamic, amortized models. Our hope is that these ideas can help lead to wider adoption of efficient, yet performant (programmable via any tolerance the user wishes to set!), models.
>
> We also note that although mathematically more formal, the calibration process in itself is remarkably simple to implement, and only requires tuning a single parameter under our framework. The overall complexity of the model, in any learning theory sense (if that is what the reviewer is referring to), is trivially affected—and effectively accounted for by the calibration process. In fact, it is this additional process itself that gives us more confidence that the method will do well in the real world—at the very least by guaranteeing the desired performance on new exchangeable inputs (which naive methods do not even afford).

---

### Official Review · Reviewer_KxKz · 2022-07-12

**Rating:** 8
**Confidence:** 4
**Soundness:** 4 excellent
**Presentation:** 3 good
**Contribution:** 4 excellent

**Summary:**

This paper proposes Confident Adaptive Language Modeling (CALM) to early-exit dynamically during the decoding process. The authors provide a theoretical guarantee that local confidence measures could lead to the confident generation of global outputs.

**Questions:**

It is surprising that the oracle early exiting only uses less than 2 layers on average. How is it possible?

There is still a large room between the oracle and the proposed methods. What are the potential directions to reduce this gap? Which assumptions can be the most easily relaxed for better efficiency?

I wonder whether the observed patterns in Figure 2 will repeat on any tasks.

Could you explain more about why the softmax-based measure gives the highest speedup compared to other measures?


**Strengths And Weaknesses:**

CALM is practically useful to speed up the inference where autoregressive decoding with large language models is quite expensive. The method is principled based on the learn-then-test framework.

---

> ### Author Response · Authors · 2022-08-01
> **Thank you**
>
> We thank the reviewer for the helpful comments. We respond to your questions below:
>
> > *It is surprising that the oracle early exiting only uses less than 2 layers on average. How is it possible?*
>
> Our oracle result is indeed very encouraging; we were excited and motivated by this potential in early experiments. This result emphasizes the potential in adaptively distributing the compute effort across tokens (compared to the static baseline of using the same fixed number of layers as the average adaptive layer, which leads to significant performance drops). Figure B.5 in the appendix provides a hint to why early-exiting is possible. As long as a select few tokens (marked in red) use the full capacity of the decoder, most of the tokens (marked in green) can be solved with a single decoding layer.
>
> That said, the oracle is still exactly that: an oracle. The oracle has access to the true consistency labels at each time step, and has the luxury of picking when to exit at exactly the right time, i.e., at exactly the first time that an intermediate layer guesses the correct output token (even if by luck). Experimentally, our hope is to get closer and closer to this performance, but we do view it as a hard upper bound.
>
> >  *There is still a large room between the oracle and the proposed methods. What are the potential directions to reduce this gap? Which assumptions can be the most easily relaxed for better efficiency?*
>
> As explained above, the oracle performance represents a perfect local confidence measure. Nevertheless, the empirical confidence measures that we propose and analyze already obtain significant efficiency gains (Figure 3), and are much closer to the optimal oracle than the static baselines.
>
> Increasing the complexity of the early exit classifier is likely to help identify earlier consistent layers with higher confidence—but of course, also with higher overhead. For example, in the limit, we could match the oracle by defining the early exit classifier as a copy of the original network, and simply check if the network’s full prediction matches the current one, though clearly the efficiency gains will be erased (rather, made up to 2x worse!). In future work we plan to study alternative ways for improving the predictive power of the exit classifier, while still keeping it fast to compute.
>
> > *I wonder whether the observed patterns in Figure 2 will repeat on any tasks.*
>
> We ran the sampling perturbations also on the WMT and Squad tasks and observed a similar pattern.
>
> > *Could you explain more about why the softmax-based measure gives the highest speedup compared to other measures?*
>
> The softmax measure allows us to directly get a signal for how strongly the predictive distribution at an intermediate layer is peaked at a single choice; i.e., it yields the margin to the next best prediction. Empirically, we find this to be a very good predictor of consistency with the final layer’s prediction, as has previous work in early exiting for NLP (e.g., [Schwartz et al., 2020](https://arxiv.org/abs/2004.07453); [Schuster et. al., 2021](https://arxiv.org/abs/2104.08803)).
>
> A possible explanation with regards to the underlying reason for its success, is that once a prediction is highly peaked on a single answer, the hidden state will have to dramatically change in order to support a different prediction (recall that we share the same softmax classifier across layers). Meanwhile, [Geva et al., 2022](https://arxiv.org/abs/2203.14680) found that layers of the Transformer act as promoters of certain concepts. Once one prediction was strongly promoted, it’s empirically less likely that other predictions will take over later (For example, see our answer to reviewer Jyd3 question: we find that 93% of the time after an intermediate layer’s prediction is consistent with the top layer, the following layers will maintain the same prediction).

---

### Meta-Review · Area_Chair_mapz · 2022-08-26

**Recommendation:** Accept
**Confidence:** Certain

**Metareview:**

This paper studies the error of early exit in decoding Transformer Language models, and proposes a method CALM to calibrate and accelerate model inference. Experiments on a variety of tasks (summarization, MT, QA) show effectiveness of the proposed method.
All reviewers find the paper solid and the author feedback convincing.

**Award:**

No

---

### Decision · Program_Chairs · 2022-09-14

Accept